# Cohort profile: the Dutch famine birth cohort (DFBC)— a prospective birth cohort study in the Netherlands

Laura S Bleker [1], Susanne R de Rooij [1], Rebecca C Painter [2], Anita CJ Ravelli [3], Tessa J Roseboom [4]

¹Epidemiology and Data Science, Amsterdam UMC, University of Amsterdam, Amsterdam, The Netherlands
²Obstetrics & Gynaecology, Amsterdam UMC, University of Amsterdam, Amsterdam, The Netherlands
³Medical Informatics; Obstetrics and Gynaecology, Amsterdam UMC, University of Amsterdam, Amsterdam, The Netherlands
⁴Epidemiology and Data Science; Obstetrics and Gynaecology, Amsterdam UMC, University of Amsterdam, Amsterdam, The Netherlands

**Correspondence to**
Dr Susanne R de Rooij;
s.r.derooij@amsterdammumc.nl

## ABSTRACT

**Purpose** The Dutch famine birth cohort study was set up to investigate the effects of acute maternal undernutrition of the 1944–1945 Dutch famine during the specific stages of gestation on later health, with a particular focus on chronic cardiovascular and metabolic diseases, ageing and mental health.

**Participants** The Dutch famine birth cohort consists of 2414 singletons born alive and at term in the Wilhelmina Gasthuis in Amsterdam around the time of the Dutch famine (1943–1947) whose birth records have been kept. The cohort has been traced and studied since 1994, when the first data collection started. The cohort has been interviewed and physically examined in several waves of data collection since that time, allowing repeated measures of a wide range of phenotypic information as well as the collection of biological samples (blood, urine, buccal swabs), functional testing (of heart, lungs, kidney, HPA axis) and imaging of the brain (MRI) and vasculature (ultrasound). Additionally, genetic and epigenetic information was collected. Through linkage with registries, mortality and morbidity information of the entire cohort has been obtained.

**Findings to date** Prenatal famine exposure had lasting consequences for health in later life. The effects of famine depended on its timing during the gestation and the organs and tissues developing at that time, with most effects after exposure to famine in early gestation. The effects of famine were widespread and affected the structure and function of many organs and tissues, resulted in altered behaviour and increased risks of chronic degenerative diseases and increased mortality. The effects of famine were independent of size at birth, which suggests that programming may occur without altering size at birth.

**Future plans** As the cohort ages, we will be assessing the effects of prenatal undernutrition on (brain) ageing, cognitive decline and dementia, as well as overall morbidity and mortality.

**Registration** The Dutch famine birth cohort is not linked to a clinical trial.

## INTRODUCTION
### Study rationale

The Dutch famine birth cohort study is a historical birth cohort of men and women born around the time of the 1944–1945 Dutch famine in the Wilhelmina Gasthuis

### Strengths and limitations of this study

► The quasi-experimental design of the study provides a unique opportunity to investigate effects of under-nutrition in pregnancy that would not be ethically possible to study otherwise.

► The long period of follow-up allows for unusually long-term consequences to be investigated decades after the exposure of interest.

► A limitation is the relatively small size of the cohort limiting the statistical power to investigate effects of prenatal famine exposure on outcomes that are rare.

► Selective survival and participation may have introduced bias in the studies which included physical examinations or measurements, but linkage to registries give insight into the degree of selection that has occurred.

► No information is available on the time period between discharge from the hospital after birth and age 50 when the cohort was first traced, leaving limitations for adjusting for potential confounders that may have had an influence during this period.

(WG) in Amsterdam, the Netherlands. The study was set up to investigate the effects of prenatal exposure to the Dutch famine on adult health in order to semiexperimentally test the developmental origins of adult disease hypothesis in a human setting. Based on this hypothesis, we expected increased rates of chronic diseases such as type 2 diabetes and cardiovascular disease among those who had been exposed to the Dutch famine prenatally.

This paper describes the rationale for the Dutch famine birth cohort study, why it was set up, how it was set up, and which unique features of the Dutch famine provided the opportunity to semiexperimentally investigate effects of prenatal undernutrition in a human setting and allow for the long-term consequences to be investigated decades later. We provide an overview of the studies that have been done in the cohort so far, the findings of 25 years of research in this cohort

and which insights they provided as well as the implications these insights might have.

## The developmental origins of health and disease

Epidemiological studies in populations across the globe have consistently shown that small size at birth is associated with increased risks of chronic degenerative diseases and their biological risk factors such as hypertension, type 2 diabetes and cardiovascular disease.[1–6] The initial observations led to the hypothesis that undernutrition during critical periods of gestation would lead to alterations in the structure and function of organs and tissues developing at the time, resulting in reduced size at birth as well as an increased risk of disease in later life.[7 8] In experimental studies in animals, undernutrition during pregnancy induced cardiovascular and metabolic changes in the adult offspring leading to increased rates of disease and reduced lifespan.[9–11] Experimentally investigating the long-term effects of undernutrition during gestation in humans would not be ethical nor practically feasible.

## The unique setting of the Dutch famine to semiexperimentally investigate effects of prenatal undernutrition in humans

The tragic circumstances of the Dutch famine of 1944–1945 created a unique opportunity to assess the effects of prenatal undernutrition on human health in later life. During World War II, part of the Dutch population was exposed to famine during the severe 'hunger winter' of 1944–1945. This historical tragedy has provided the unique opportunity to study short and long-term effects of famine exposure during intrauterine development. The Dutch famine can be viewed as a 'natural experiment', because of its sudden onset and end in a previously well-nourished population. The famine was acute, relatively short and had an impact on the entire population within the affected regions, minimising confounding by genetic inheritance or other simultaneously occurring risk factors, such as socioeconomic status. Also, throughout the famine, food rations were precisely registered.

## Historical events leading to the Dutch famine

In 1944–1945, the Dutch famine hit the western parts of the Netherlands towards the end of World War II, only months before its liberation from the German occupier by the Allied forces. Before the war, nutritional standards of the Dutch population were generally adequate.[12] On the 10 May 1940, the Netherlands were invaded by German forces, with immediate impact on the Dutch food supply. Food imports from other countries, including the Dutch colonies, were no longer possible, and part of the food produced in the Netherlands was sent to Germany.[12] The National Bureau for Food Distribution had already implemented food rationing and distribution prior to the invasion, and was fully operative throughout the entire country by the end of April 1941. The food rations were determined weekly and recorded in detail. However, daily rations remained calorically and nutritionally adequate until September 1944, when they suddenly dropped below 1600 calories.[13 14] On the 17 September 1944, the Dutch exiled government requested a railroad strike to support the advance of the Allied forces, which by that time had already liberated the southern part of the Netherlands. The German occupier responded by an abrupt ban of food transports to the western part of the Netherlands. By November 1944, food stocks in the large cities in the west of the Netherlands had been exhausted, and the ban was partly lifted to allow food transport across water. However, due to an early and exceptionally harsh winter, waterways and canals had frozen over. Food stocks could no longer be replenished and famine set in for the inhabitants of the large cities in the west, including Amsterdam, Rotterdam and The Hague. The official daily rations per person dropped to below 1000 calories in November 1944 and varied between 400 and 800 calories in the following 6 months, a period that would later be referred to as the 'Hunger Winter'. Children aged 1 year or younger were relatively protected, as their official daily ration never dropped below 1000 calories. Although this initially also applied to pregnant and lactating women, this was no longer possible as the food scarcity persisted. On the 5 May 1945, the western parts of the Netherlands were liberated by The Allied Forces. Within a week, daily rations had rapidly risen above 1000 calories, through supplements by airdrops and the Special Red Cross Feeding Team.[15] By June 1945, rations had risen above 2000 calories. Within weeks, the prefamine nutritional standards were re-established, and were maintained until the food ration was permanently lifted, 5 years after the war in 1950.[16 17]

## STUDIES OF THE DUTCH FAMINE

Soon after the war ended, researchers realised the unique setting the Dutch famine had unintentionally created to semiexperimentally investigate effects of famine exposure in pregnancy on offspring's health. Since, several research groups have investigated the long-term consequences of prenatal exposure to the Dutch famine. These studies have used a variety of different designs; register studies have linked date and place of birth to various outcomes, data from recruitment tests of military conscripts have been used to study links between prenatal famine exposure (based on time and place of birth) and IQ, mental health and body mass index (BMI), and birth records have been used to set up cohorts comparing those exposed to famine prenatally with their unexposed same-sex siblings, or to those born in the same hospital either before the famine or conceived after it. Some of the earliest studies on the Dutch famine were published within a few years after World War II by Smith and Sindram, and showed that babies born during the famine were lighter at birth.[18 19]

In the 70s, data from 19-year old Dutch military conscripts were used to investigate effects of prenatal famine exposure on IQ[17] and BMI, showing no effects on intelligence but an increased prevalence of obesity

among military conscripts who were exposed to famine during early gestation.[20 21]

Using the Dutch Psychiatry Registry, increased rates of schizophrenia,[22 23] and affective disorders[24 25] were observed in those prenatally exposed to famine. Refined analyses of the military conscripts data, later confirmed these findings and showed higher rates for schizophrenia,[26] a schizoid personality[27] and antisocial personality disorder,[28] in men who had been exposed to the Dutch famine prenatally.

Barkers findings of links between small size at birth and later cardiovascular and metabolic disease and the hypothesis that fetal undernutrition would underlie this association[29] led to a collaborative effort between the Academic Medical Center in Amsterdam (the successor of the WG that had kept birth records of babies born around the time of the famine) and the Medical Research Council (MRC) Environmental Epidemiology Unit in Southampton, UK, led by Barker, to set up the Dutch famine birth cohort study.

## COHORT DESCRIPTION
### Eligibility
In 1994, the Dutch famine birth cohort was set up in the Academic Medical Centre in Amsterdam, the Netherlands. All men and women born as term singletons between 1 November 1943 and 28 February 1947 in the WG in Amsterdam were eligible candidates to be included in the Dutch famine birth cohort. The WG was the main maternity hospital in Amsterdam at the time of the famine and birth records were still kept at the city archive of Amsterdam.[30] Only live born singletons after a full-term pregnancy were included (≥259 days gestational age, calculated either from the date of the last menstrual period or by the obstetrician's estimation at first prenatal visit and at physical examination of the child at birth) in order to exclude the potential confounding effects of prematurity or twinning on later health.

### Study participants
The birth ledgers, which contained general information on date of birth, name of the parents and identification number were retrieved from the Obstetrics department of the Academic Medical Center (the former WG) in Amsterdam of individuals born between 1 November 1943 and 28 February 1947. From the 5425 records, all individuals exposed to famine, according to our definition (see below), were considered eligible exposed candidates. A random sample of eligible individuals unexposed to famine during gestation (born before the famine or conceived after the famine) was taken. Twins and stillbirths were excluded. This resulted in 2680 eligible candidates for which the detailed medical records were searched and coded in all detail in the city archive of Amsterdam. Of these 2680 individuals, 27 individuals (1.0%) were excluded because their medical records were missing or incomplete, and 239 individuals (8.9%) were excluded

based on their preterm birth (<259 days gestational age). Ultimately, 2414 people were considered eligible (821 live born term singletons exposed to famine prenatally, 764 live born term singletons born before the famine and 829 live born term singletons conceived after the famine).

### Exposure to famine
Exposure to famine during gestation was defined as an average maternal daily ration of less than 1000 calories during any 13-week period of gestation.[31] According to the official daily rations of the general population, children born in Amsterdam between 7 January 1945 and 8 December 1945, were considered to be exposed to famine during prenatal life. Three 16-week periods were distinguished; children who were mainly exposed during late gestation (born between 7 January and 28 April 1945), mid gestation (born between 29 April and 18 August 1945) or early gestation (born between 19 August and 8 December 1945). A sample of the individuals born within 1 year before the famine or conceived up to 1 year after the end of the famine were eligible for controls as comparisons. Being unexposed to famine was defined as being born before the famine (born between 1 November 1943 and 7 January 1945), or conceived after the famine (born between 9 December 1945 and 28 February 1947) (figure 1).

### Maternal and birth characteristics
Medical records contained information on maternal characteristics including maternal weight and blood pressure throughout pregnancy as well as a detailed description of delivery and the postpartum period.

Maternal characteristics included maternal age, marital status, reproductive and medical history, and occupation of the head of the family. Blood pressure and maternal weight were measured at every prenatal visit. Pelvic measurements were taken, including the interspinous distance (the distance between the anterior superior iliac spines). Maternal weight gain in the third trimester was estimated as the difference between the woman's weight as close as possible to the start of the third trimester and her weight at the last prenatal visit (always within 2 weeks of birth), multiplied by the ratio of trimester duration (13 weeks) and the time interval between the two weight measurements.

Birth characteristics included date of birth, sex, birth weight and birth length (crown to heel). Head circumference was calculated by: $\varpi$ * (biparietal diameter +occipitofrontal diameter) * 0.5. Ponderal index was calculated by: birth weight/(crown-to-heel$^2$). Placental area was calculated by: $\varpi$ * placental length * placental width * 0.25. At discharge, infant weight and type of feeding (exclusive breastfeeding/exclusive bottle feeding/a combination) were recorded. Besides information from the medical records for each woman the profession of the father/the head of the family was collected using the hospital admission 'crib cards' and coded as manual or non-manual work. Table 1 shows the maternal and birth

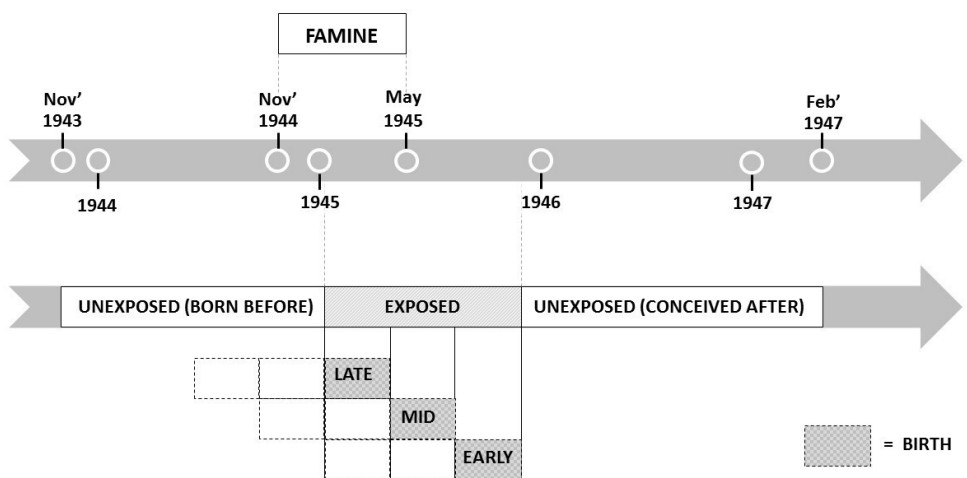

**Figure 1** Dutch famine birth cohort schematic presentation of gestational famine exposure.

characteristics of the cohort according to the timing of exposure to famine.

### Follow-up

In 1994, current addresses of the children born before, during and after the famine were requested from the population registry (Bevolkingsregister) in Amsterdam, which traced 2155 of the 2414 eligible participants. Of these, 265 persons had died, 199 had emigrated and another 164 refused to share their address, resulting in 1527 (63.3%) of 2155 cases. Power calculations for a study sample to be invited for interviewing and clinical measurements were based on the 120 min glucose concentrations after a 75 g

oral glucose load as the main outcome variable, based on pooled data from three previous MRC Southampton studies which had established a link between birth weight and glucose tolerance in adulthood (Hertfordshire, Preston and Sheffield).[32] To detect an increase of 10% in the 120 min glucose concentration, a sample size of 700 men and women (450 prenatally exposed and 250 unexposed persons) was needed for 93% power and a 5% type 1 error probability (two sided). A 70% participation rate for the oral glucose load test was assumed and based on this, interviews with 1000 men and women at their homes were planned. We invited all people in each of the five

**Table 1** Maternal characteristics and birth outcomes of 2414 singleton children born alive and at term in the Wilhelmina Gasthuis in Amsterdam between 1 November 1943 and 28 February 1947

| | | Exposure to famine | | | | | |
|---|---|---|---|---|---|---|---|
| | Born before n=764 | Late gestation n=307 | Mid gestation n=297 | Early gestation n=217 | Conceived after n=829 | Total (SD) n=2414 | Missing (N) |
| Maternal characteristics | | | | | | | |
| Age (years) | 28.5 | 30.0 | 28.1 | 27.7 | 28.0 | 28.3 (6.4) | 0 |
| Never married (%) | 13.2 | 9.8 | 20.2 | 25.8 | 16.3 | 15.8 | 0 |
| Weight gain third trim. (kg) | 3.1 | 0.0 | 4.9 | 5.7 | 4.2 | 3.5 (3.2) | 732 |
| Weight at last prenatal visit (kg) | 66.7 | 61.8 | 63.5 | 67.9 | 69.1 | 66.6 (8.7) | 281 |
| Interspinous distance (cm) | 26.1 | 25.8 | 25.5 | 25.8 | 26.3 | 26.0 (1.8) | 14 |
| Birth outcomes | | | | | | | |
| Boys (%) | 53.1 | 48.2 | 48.5 | 49.8 | 52.5 | 51.4 | |
| First born (%) | 39.7 | 29.6 | 36.7 | 39.2 | 39.3 | 37.9 | |
| Pregnancy duration (days) | 285 | 283 | 285 | 287 | 286 | 285 (11) | 371 |
| Birth weight (g) | 3373 | 3133 | 3217 | 3470 | 3413 | 3346 (478) | 0 |
| Body length (cm) | 50.5 | 49.4 | 49.8 | 50.9 | 50.5 | 50.3 (2.2) | 32 |
| Ponderal index (kg/m$^3$) | 26.1 | 25.8 | 26.0 | 26.2 | 26.5 | 26.4 (2.4) | 32 |
| Head circumference (cm) | 32.9 | 32.3 | 32.1 | 32.8 | 33.2 | 32.8 (2.4) | 17 |
| Placental diameter (cm) | 20.6 | 19.7 | 20.0 | 19.9 | 20.2 | 20.2 (2.5) | 357 |

*Adapted from table 1, page 59, Ravelli.[58]

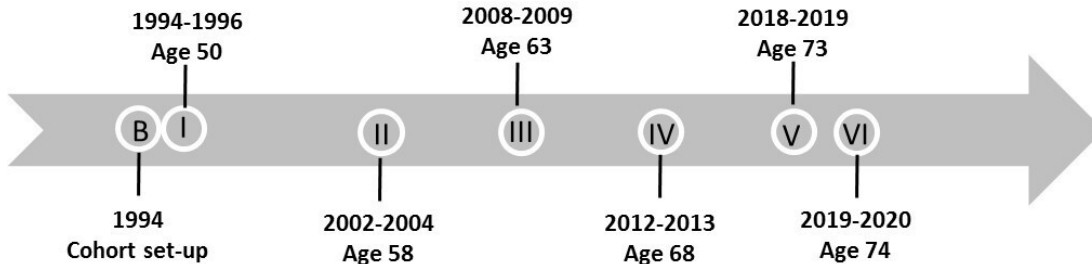

**Figure 2** Dutch famine birth cohort schematic presentation of study waves.

exposure/non-exposure groups, starting with those who lived in, or close to, Amsterdam (n=1018) to participate in the first study, of whom 912 (90%) agreed to be interviewed at their homes and 741 (81%) of the interviewed participants visited the clinic for measurements.[31]

Data collection of the Dutch famine birth cohort study has occurred in six study waves from 1994/1996 to 2020, with a seventh wave planned in 2021 (see figure 2 for an overview of study waves). Measurements consisted of a home interview at age 50 and a hospital visit around the ages of 50 (wave I, 1994–1996), 58 (wave II, 2002–2004), 63 (wave III, 2008–2009), 68 (wave IV, 2012–2013) and 74 (wave VI, 2019–2020) and completion of a questionnaire by the participants at age 73 (wave V, 2018). At wave V in 2018, 1207 of the 1527 participants were still living in the Netherlands with known addresses (79%), of which 595 agreed to participate (49%). The fourth and sixth waves occurred in subsamples of the cohort.

## Measurements
Table 2 gives a summary of the study waves, including the mean ages of the famine exposed participants during that wave, with outcome measures and a description of participation rates. Outcome measures were compared between those exposed to famine in early, mid or late gestation to those prenatally unexposed to famine, mostly by applying regression models with exposures as dummy variables.

## Patient and public involvement statement
Cohort members are informed about study aims and outcomes via newsletters and the studies website www.hongerwinter.nl. Cohort members have not been involved in setting up research questions and data collection, study design, recruitment strategies, assessment of study burden and dissemination of study results.

## Findings to date
Prenatal exposure to undernutrition appears to have (1) permanent effects on human health. The effects of undernutrition (2) depend on its timing during gestation and the organs and systems developing during that critical time window. The effects on later health were most pronounced among those exposed to famine in early gestation. This may not be surprising considering the fact that all organs are formed in early gestation and insufficient food supply during the formation of the organs interferes most with future physiological

functions. The effects of famine were independent of size at birth (3) as there were effects of prenatal famine exposure in the absence of effects on body size at birth. This implies that adaptations that enable the fetus to grow can nevertheless have adverse long-term consequences.

Below, we describe the findings according to the three main conclusions.

### Prenatal famine exposure has lasting effects on health
Table 3 gives an overview of findings in the Dutch famine birth cohort study. Exposure to famine was associated with a threefold higher risk for[33] and earlier onset of coronary heart disease (CHD),[34] a more atherogenic lipid profile,[35] glucose intolerance,[31 36] a disturbed blood coagulation,[37] microalbuminuria and a decrease in creatinine clearance,[38] as well as more respiratory complaints and obstructive airways disease in adulthood.[39] Also, those exposed to famine during gestation were more likely to consume a high-fat diet,[40] and more often perceived their own health as 'poor'.[41] Although there was no indication that prenatal undernutrition increased resting blood pressure,[42] those exposed to undernutrition in utero had a higher increase in systolic blood pressure in response to a stressful assignment compared with unexposed individuals.[43] Moreover, they performed worse on a selective attention task, a cognitive ability that usually declines with increasing age.[44] Women prenatally exposed to famine were more centrally obese,[45] had a five times increased risk for developing breast cancer[46] and an overall higher mortality rate, as well as a higher cardiovascular mortality, cancer mortality and breast cancer mortality.[47] In men, exposure to famine during gestation was associated with smaller intracranial volume.[48] Moreover, their brain appeared older, as assessed by a machine-learning pattern recognition method that is able to estimate individual brain ages based on T1-weighted MR images, and overall perfusion of the brain was worse compared with controls.[49 50] Also, prenatal exposure to famine was associated with increased symptoms of depression and anxiety,[51] and a lower physical performance score in men.[52] In terms of transgenerational effects, babies born to women exposed to famine prenatally were more adipose as neonates, and parents that had been exposed to famine in utero themselves more often reported their child to have a poorer health.[53] Offspring of men exposed

**Table 2** Dutch famine birth cohort study waves, including the mean ages of the participants at examination, the outcome measures and a description of attrition rates

| Phase | Measurement | Lost to follow-up |
|---|---|---|
| Wave I 1994–1996 | Wave 1a birth data collection/exposure definition/mortality status. | |
| Wave I (age 50) 1994–1996 | Wave Ib (home interview)<br>▶ Questionnaires (General information, medical information, life style factors, physical activity, weight history, reproductive history, EPIC Food frequency questionnaire, self-perceived health and medication use).[41 68]<br>▶ Anthropometrics.[45]<br>▶ Blood pressure.[42 69]<br>Wave Ic (hospital visit)<br>▶ Blood pressure.[42 69]<br>▶ Glucose tolerance test.[31]<br>▶ ECG.[33]<br>▶ Lung function.[39]<br>▶ Urine collection.[38]<br>▶ Lipid profile and clotting factors.[35 37] | 912 (90%) of the 1018 selected participants were interviewed at their homes (Wave Ib), 741 (81%) agreed to visit the AMC clinic (Wave Ic). There were no differences in the mean birth weight of the 741 who attended the clinic and the 2414 children in the original cohort. |
| Wave II (Age 58) 2002–2004 | Wave IIa (hospital visit)<br>▶ Questionnaires (General information, medical information, lifestyle factors, physical activity, weight history reproductive history, transgenerational effects, self-perceived health and medication use).[40 46 51 53 70–73]<br>▶ Glucose tolerance test.[36 74]<br>▶ Anthropometrics.<br>▶ Blood pressure.<br>▶ ECG.[34]<br>▶ Urine collection.[38]<br>▶ Cognitive function.[44]<br>▶ Ultrasound examinations of the arterial walls of the carotid and femoral arteries.[75 76]<br>▶ Physical function.[77]<br>▶ Psychological Stress tests.[43 78 79]<br>▶ Genomic DNA from blood plasma.[80–84]<br>Wave IIb (hospital visit)<br>▶ Intravenous glucose tolerance test in a subsample (n=94).[85]<br>Wave IIc (hospital visit)<br>▶ Synacthen test in a subsample (n=98).[86] | 860 (60%) of 1423 eligible candidates agreed to participate. There were no differences in mean birth weight between responders and non-responders. |
| Wave III (age 63) 2008–2009 | ▶ Questionnaires (General information, medical information, lifestyle factors, physical activity, weight history reproductive history, transgenerational effects, self-perceived health and medication use).[87–89]<br>▶ Transgenerational effects based on F2 questionnaire (General information, birth characteristics, self-perceived health, exercise, medical information, lifestyle factors).[54]<br>▶ F0-F1-F2 (grandmother-parent-child) buccal swab for DNA methylation. | 601 (44%) of 1372 eligible candidates agreed to participate. 483 F2s (grandchildren) were willing to participate, of which 360 (74.5%) completed the questionnaire. Birth weight or gestational age did not differ between F1 participants and F1 non-participants. |
| Wave IV (age 68) 2012–2013 | ▶ Questionnaires (General information, medical information, lifestyle factors, physical activity, weight history reproductive history, transgenerational effects, self-perceived health and medication use).<br>▶ Anthropometrics.<br>▶ Blood pressure.<br>▶ Glucose concentration (non-fasting).<br>▶ Lipid profile (non-fasting).<br>▶ Cognitive function.<br>▶ Brain imaging (MRI) (white matter hyper intensities, cerebral micro bleeds, total cortical, hippocampal and lacunar volume, brain perfusion, resting brain state conditions, Brain age, brain perfusion).[48–50]<br>▶ Ageing parameters (Stroke incidence, fractures and osteoporosis, physical performance, visual acuity, cataract, hearing).[52]<br>▶ Cellular ageing (telomere length).[90] | Equal samples were drawn randomly from each of the groups (1307 eligible candidates in total) until the number of 50 people per exposure group (born before famine, exposed in early gestation, conceived after famine) agreeing to participate was reached. A total number of 151 participants of an eligible group of 268 cohort members (56%) were visited at home. There was a difference in birth weight between the participants (n=150) and the nonparticipants (n=1157) of the eligible group (n=1307), 3438 vs 3346 g; p=0.03. |

**Table 2** Continued

| Phase | Measurement | Lost to follow-up |
|---|---|---|
| Wave V (age 73) 2018–2019 | ▶ Questionnaires (including self-perceived health, daily life functioning, pain complaints, mood, memory, attention, and cognition, diseases, tasks and activities in daily life, social activities, quality of life, medical care consumption, stressful events, health problems resulting from stressful events, and childhood experiences). | 595 (49%) of 1207 eligible candidates agreed to participate. Data analyses are currently ongoing. |
| Wave VI (age 74) 2019–2020 | ▶ Questionnaires (General information, medical information, lifestyle factors, physical activity, weight history reproductive history, transgenerational effects, self-perceived health and medication use).<br>▶ Anthropometrics.<br>▶ Glucose (non-fasting).<br>▶ Lipid profile (non-fasting).<br>▶ Blood pressure.<br>▶ Cognitive function.<br>▶ Brain Imaging (MRI) (white matter hyper intensities, cerebral micro bleeds, structural total and area brain volumes, brain perfusion, BrainAge, resting brain state conditions, active brain state conditions during Stroop selective attention task). | Cohort members who underwent MRI scanning in Wave IV were re-invited for this study. A total of 92 participants were seen in this study. Data analyses are currently being set up. |

AMC, Academic Medical Centre; EPIC, European Prospective Investigation into Cancer and Nutrition.

to famine in utero had an increased weight and a higher BMI at age 37 years.[54]

### The effects were dependent of its timing during gestation (and the organs and tissues growing at that time)

Our findings suggest that famine exposure during periods of rapid organ growth permanently affects organ structure. The majority of adverse effects of prenatal famine on several (precursors of) chronic diseases were observed specifically in those exposed during early gestation.[33–35 37 40 41 43–52] Since organogenesis takes place in early gestation it is not surprising that exposure to undernutrition during this critical period of development has the most detrimental effects on later life physical and mental health. The effects on renal function and respiratory complaints were observed only in those who had experienced famine in utero during mid gestation.[38 39] This fits with the stage of rapid expansion of nephron number and of rapid bronchial tree branching in mid-pregnancy in fetuses.[55 56] Finally, across all trimesters, but particularly in mid and late gestation, famine was associated with offspring glucose intolerance in later life.[31 36] During mid and late gestation, the endocrine pancreas undergoes rapid proliferation.[57] Altogether, our findings are highly suggestive of specific windows of increased sensitivity for negative effects of famine on various organ systems, depending on the gestational period in which cell division is at its peak.

### The effects were independent of size at birth

One of the findings of the Dutch famine birth cohort study is that babies born after exposure to the famine in mid (difference = −165 g, 95% CI −220 to −109) or late (difference=−257 g, 95% CI −312 to −202) gestation had lower birth weights, whereas babies born after exposure to the famine in early (difference=79 g, 95% CI 16 to 143) gestation had higher birth weight compared with controls (table 1).[58] This is tremendously striking, since the long-term programming effects of prenatal maternal undernutrition on adult health have traditionally been assumed to be mediated by poor fetal growth and hence marked by a low birth weight. However, we found that the majority of the effects of prenatal famine exposure on adverse health outcomes in adults were observed in those exposed to famine in early gestation, and thus, in those with normal birth weight. Moreover, additional adjustment for birth weight showed that the effect of prenatal exposure to famine on the adult physical and mental health outcomes were independent of birth weight. In other words, prenatal famine had a major negative affect on adult health, without affecting birth weight. Our findings, thus, suggest that adverse effects of maternal famine exposure on fetal organ development and function are independent of size at birth.

Further information, including a list of publications can be found on the cohort's web page (www.hongerwinter.nl).

### Strengths and limitations

A strength of the Dutch famine birth cohort study is its quasi-experimental design. This design reduces the chance that associations between prenatal famine exposure and adult health outcomes are due to genetic inheritance of mother and child or socioeconomic status, since exposure to famine was completely independent of maternal characteristics such as age or parity, socioeconomic status and other environmental circumstances that may affect long-term health of their children. Nevertheless, exposed and unexposed individuals may have differed on other genetically inherited characteristics which we did not measure. The Dutch hunger

**Table 3** Adult characteristics of the Dutch famine birth cohort study from the age of 50 years, divided in columns based on the gestational period of prenatal famine exposure

| Age | Ref | Adult characteristics | Born before | Late gestation | Mid gestation | Early gestation | Conceived after | Mean (SD) | N |
|---|---|---|---|---|---|---|---|---|---|
| **50** | 33 | Coronary heart disease (%) | 3.8 | 2.5 | 0.9 | **8.8§** | 2.6 | 3.3 | 736 |
| | 35 | LDL:HDL cholesterol* | 2.91 | 2.82 | 2.69 | 3.26 | 2.94 | 2.90 (1.53) | 704 |
| | 31 | 120 min. Glucose* (mmol/L) | 5.7 | **6.3§** | **6.1§** | **6.1§** | 5.9 | 6.0 (1.4) | 702 |
| | | 120 min. insulin* (mmol/L) | 160 | **200§** | **190§** | **207§** | 181 | 181 (2.4) | 694 |
| | 37 | Fibrinogen (g/l) | 3.02 | 3.05 | 3.05 | 3.21 | 3.10 | 3.07 (0.6) | 725 |
| | | Factor VII* (% of standard) | 127.7 | 130.7 | 132.7 | **116.9§** | 132.6 | 129.4 (1.4) | 725 |
| | 42 | SBP (mm Hg) | 126.0 | 127.4 | 124.8 | 123.4 | 125.1 | 125.5 (15.5) | 739 |
| | | DBP (mm Hg) | 86.2 | 86.4 | 84.4 | 84.8 | 85.2 | 85.6 (9.9) | 739 |
| | 45 | BMI* (kg/m†) | 26.7 | 26.7 | 26.6 | **28.1¶§** | 27.2 | 27.0 (1.2) | 741 |
| | | Waist circumference (cm) | 91.8 | 92.4 | 91.0 | **95.6¶§** | 92.5 | 92.3 (13) | 741 |
| | 41 | Perceived health poor (%) | 4.5 | 6.4 | 3.7 | **10.3§** | 5.3 | 5.5 | 912 |
| | 38 | Microalbuminuria (ACR 2.5) (%) | 8 | 7 | **12§** | 9 | 4 | 7 | 724 |
| | 39 | FEV1/FVC (x100) | 0.72 | 0.72 | 0.72 | 0.72 | 0.74 | 0.72 (0.1) | 726 |
| | | Obstructive airways disease (%) | 15.5 | 15.0 | **24.8§** | 23.0 | 17.3 | 18.1 | 726 |
| | | Total IgE* (IU/mL) | 31.0 | 27.2 | 28.9 | 27.9 | 33.1 | 30.4 (4.6) | 726 |
| | 91 | Mortality 18–50 years (%) | 5.3 | 3.9 | 2.8 | 5.0 | 3.6 | – | 2254 |
| **58** | 43 | ΔSBP (mm Hg) Stroop response | 19 | 20 | 19 | **23§** | 19 | 20 (16) | 721 |
| | 36 | 120 min. glucose* (mmol/L) | 5.8 | **6.2§** | **6.2§** | **6.2§** | 5.9 | 6.0 | 678 |
| | | 120 min. insulin* (pmol/L) | 242 | **263§** | **254§** | **269§** | 240 | 249 (2.1) | 672 |
| | 40 | >39% of energy from fat (%) | 25.9 | 28.7 | 20.2 | **40.7§** | 24.9 | – | 730 |
| | 44 | Stroop Task Score† (% correct) | 42.3 | 36.5 | 40.0 | **27.5§** | 43.9 | 38.5 (55.7) | 714 |
| | | AH4 test Score† (% correct) | 70.9 | 72.4 | 71.8 | 76.0 | 73.3 | 72.4 (19.0) | 727 |
| | | Memory task retrieval (%) | 81.8 | 78.5 | 83.2 | 79.7 | 79.6 | 80.7 (19.9) | 583 |
| | | Mirror task errors per round† | 13 | 12 | 10 | 14 | 10 | 12 (30) | 643 |
| | 46 | Breast cancer (%) | 2.8 | 3.7 | 3.9 | **8.7§** | 0.8 | 3.2 | 475 |
| | 92 | Mortality 18–57 years (%) | 9.1 | 4.4 | 6.0 | 6.5 | 5.9 | – | 2245 |
| | 51 | HADS-A score | 4.5 | 4.7 | 5.1 | 5.8 | 5.1 | 4.9 (3.1) | 639 |
| | | HADS-A ≥8 points (OR) | 1.0 (ref) | 0.7 | 1.1 | **2.7**§ | 1.0 | – | 369 |
| | | HADS-D | 3.1 | 3.0 | 3.1 | **4.9**§ | 3.6 | 3.2 (3.1) | 369 |
| | | HADS-D ≥8 points (OR) | 1.0 (ref) | 0.9 | 0.6 | 1.7 | 1.0 | – | 369 |
| **63** | 47 | Overall adult mortality (HR) | 1.0 (ref) | 0.8 | 0.5 | **1.9¶§** | 1.0 (ref) | – | 1125 |
| | | Cardiovascular mortality (HR) | 1.0 (ref) | - (zero deaths) | 0.6 | **4.6¶§** | 1.0 (ref) | – | 1125 |
| | | Cancer mortality (HR) | 1.0 (ref) | 1.0 | 0.5 | **2.3¶§** | 1.0 (ref) | – | 1125 |
| | | Breast cancer mortality (HR) | 1.0 (ref) | 2.1 | 1.5 | **8.3¶§** | 1.0 (ref) | – | 1125 |
| **68** | 48–50 | ICV (mL) | 1138 | – | – | **1101**§ | 1176 | 1138 (85) | 52 |
| | | BrainAGE score (years) | −1.81 | – | – | **2.53**§ | 0.53 | – | 52 |
| | | Spatial Coefficient of Variance of a CBF image | 0.60 | – | – | **0.64§** | 0.59 | 0.61 (0.08) | 50 |
| | 52 | Grip strength (kg) | 46.3³ | – | – | **42.1§** | 46.3³ | – | 67 |
| | | Physical performance score (points) | 9.7³ | – | – | **8.9§** | 9.7³ | – | 67 |

**Transgenerational - F2 offspring of F1¶**

| Age | Ref | Adult characteristics | Born before | Exposure to famine | | | Conceived after | Mean (SD) | n |
|---|---|---|---|---|---|---|---|---|---|
| **33** | 53 | Birth weight (grams) | 3476 | 3484 | | | 3468 | – | 856 |
| | | Ponderal index (kg/m³) | 26.6 | **27.8§** | | | 26.5 | – | 856 |
| | | F2 poor health‡ (%) | 16 | 18 | | | 14 | – | 856 |

Continued

**Table 3** Continued

| Age | Ref | Adult characteristics | Born before | Late gestation | Mid gestation | Early gestation | Conceived after | Mean (SD) | N |
|---|---|---|---|---|---|---|---|---|---|
| Age | Ref | Adult characteristics | Unexposed to famine | Exposed to famine | | | | Mean (SD) | n |
| 35 | 54 | Adult weight (kg) | 78.9 | 79.1 | | | | 79.0 | 209 |
| | | Adult BMI* (kg/m†) | 25.7 | 25.0 | | | | 25.3 | 209 |
| **Transgenerational - F2 offspring of F1**** | | | | | | | | | |
| Age | Ref | Adult characteristics | Born before | Exposure to famine | | | Conceived after | Mean (SD) | n |
| 30 | 53 | Birth weight (grams) | 3304 | 3298 | | | 856 | – | 640 |
| | | Ponderal index (kg/m³) | 25.9 | 25.9 | | | 26.3 | – | 640 |
| | | F2 poor health‡ (%) | 12 | 12 | | | 8 | – | 640 |
| Age | Ref | Adult characteristics | Unexposed to famine | Exposed to famine | | | | Mean (SD) | n |
| 35 | 54 | Adult weight (kg) | 73.5 | 78.8§ | | | | 75.3 | 151 |
| | | Adult BMI* (kg/m†) | 23.8 | 25.2§ | | | | 24.3 | 151 |

*Geometric means (SD).

†Medians (IQR).

‡Congenital: asphyxia, developmental delay, Down syndrome, congenital heart disorders; cardiovascular and metabolic: diabetes, acquired cardiovascular conditions, obesity; psychiatric: schizophrenia, depression, suicide (attempt), drug/alcohol dependency; other: accidental, acquired neurological, autoimmune, respiratory, infectious, neoplastic, dermatological conditions.

§P≤0.05 compared with unexposed (born before and conceived after the famine).

¶Measured both in men and women, results shown here only include sex-specific significant associations (women).

**Measured both in men and women, results shown here only include sex-specific significant associations (men).

BMI, body mass index; CBF, cerebral blood flow; DBP, diastolic blood pressure; FEV1/FVC, forced expiratory volume/forced vital capacity; HADS-A, Hospital Anxiety and Depression Scale-Anxiety subscale; HADS-D, Hospital Anxiety and Depression Scale-Depression subscale; ICV, intracranial volume; LDL:HDL, low density lipoprotein/high density lipoprotein; SBP, systolic blood pressure; TBV, total brain volume.

winter families study is a different cohort study that used an approach with subtle differences to that of the Dutch famine birth cohort.[59] This cohort was set up in 2003 to study adult health in individuals born in one of three institutions in famine-exposed cities (Amsterdam, Rotterdam, Leiden) to women who had been exposed to famine during or immediately preceding pregnancy. The Dutch hunger winter families study differed from the Dutch famine birth cohort study in one essential aspect: Controls were unexposed same-sex siblings and not unrelated time controls born before or conceived after the famine, to account for any potential genetic influences on later life health outcomes.[59] Interestingly, the findings between the two cohorts are strikingly similar, and suggest that the consequences that both studies find of prenatal famine exposure are due to a causal effect of famine exposure rather than due to confounding.

Another strength of our study design was that, due to the relatively short period of the famine, the exposed individuals are unlikely to substantially differ from unexposed individuals in terms of general socioenvironmental conditions, indicating that famine exposure is one of the single remaining factors that can explain differences between both groups. Factors which we cannot control for are, first, the possible heightened levels of psychological and/or physical stress in women experiencing famine, which may have affected fetal development also, or, second, specific dietary changes that co-occurred at the time of the famine which may have impacted fetal development. Examples of the latter could include tulip bulb consumption. Our analyses do not allow us to dissect effects of lack of food from stress or increased rates of infection due to famine exposure. Interestingly, the effects of postnatal famine exposure which has occurred among those born before the famine seem to be limited as the adult health outcomes of those born before the famine and those conceived after it are strikingly similar.

A relative weakness of the cohort is possible selective conception and survival. During the famine, fertility rates decreased by approximately 50%, due to increased amenorrhoea among women.[16 17] Also, perinatal mortality was higher during the famine.[17 60] Both the decreased conception rates and increase in perinatal mortality during the famine may have led to selective births of 'stronger' individuals. However, when we compared maternal characteristics such as socio-economic status, age and parity between those who conceived during the famine and those who gave birth before or conceived after the famine, we found no major differences except that early exposed women were more frequently unmarried. An increase in infant mortality of those with poorer health during the famine is also not likely to explain our findings, as the majority of the associations between prenatal famine and adverse health outcomes was observed in those conceived during the famine.

Another weakness of the study especially as the cohort grows older, is that more resilient individuals may remain alive and in good health, in particular those who are motivated to participate in the study. Selective attrition may have particularly affected the group of exposed

individuals, as we have shown that this group was particularly affected by famine exposure in utero, reflected by more deaths (in women) and poorer health outcomes. This asymmetric loss of adults with poorer health may have led to an under estimation of the true effects of prenatal famine on adult health outcomes.

A final limitation to the study is that we have no information on the time period between discharge from the hospital after birth and age 50 when the cohort was first traced, making it impossible to adjust for potential confounders that may have had an influence during this period.

### Similar findings in different settings

Studies in other settings, of famines with different durations and severity support these findings and suggest that the results of studies on the Dutch famine are not uniquely linked to the Dutch famine situation, but rather reflect biologically fundamental processes of human plasticity.[61] A study in Nigeria showed that people who had been exposed to the Biafran famine (1967–1970) prenatally had increased rates of hypertension and type 2 diabetes.[62] Similarly, studies of the Great Leap Forward famine in China have shown similar effects of prenatal famine exposure on later risks of diabetes, hypertension and schizophrenia.[63–65] And similarly studies of Austrian[66] and Ukrainian famines[67] showed increased risks of diabetes among those who had been exposed to famine prenatally.

### Implications

Studies of the Dutch and other famines show that those faced with undernutrition during the critical earliest stages of development have increased rates of chronic generative disease in adult life. These findings teach us the fundamental importance of a good start in life. Adequately feeding women before and during the pregnancy will allow future generations to reach their full potential and lead healthier and more productive lives, ultimately leading to a healthier and more equal future.

### Collaboration

The Dutch famine birth cohort study welcomes opportunities for collaboration; Enquiries should be directed to the principal investigator (t.j.roseboom@amsterdamumc.nl).

**Acknowledgements** We would like to thank the members of the Dutch famine birth cohort for their participation in our studies. We would also like to thank all researchers and other staff who have worked with us on study design, data collection, assessments, analyses and drafting of manuscripts.

**Contributors** ACJR, RP, SRdR and TJR contributed to parts of the development of methods and parts of the data collection. All authors were involved in data analysis and interpretation. LB, SRdR and TJR drafted the work. All authors have critically revised this article and approved the final version to be published.

**Funding** The Dutch famine birth cohort study has been funded by the Diabetes Fonds (The Netherlands, Grant Number NA), the Netherlands Heart Foundation ((NHS2001B087, NHS2007B083), The European Science Foundation (EUROSTRESS-DOME project), the European Commission (Brainage (Seventh Framework Programme Project 279281), Dynahealth (Horizon 2020 Project 633595), Longitools (Horizon 2020 Project 874739), Well-being (UK, Grant Number NA), the Medical Research Council (UK, Grant Number NA), the Dutch Research Council (NWO Aspasia Project 015014039) and the Academic Medical Centre (Amsterdam, The Netherlands, Grant number NA). We declare no conflict of interest.

**Patient and public involvement** Patients and/or the public were not involved in the design, or conduct, or reporting, or dissemination plans of this research.

**Patient consent for publication** Not required.

**Ethics approval** Ethical approvals for the study have been provided by the Ethics Committee of the Academic Medical Center Amsterdam and have been registered under the numbers: MEC 01/215, MEC 08/26, MEC 2011/ 305, W17_025 and MEC 2019 / 017.

**ORCID iDs**
Laura S Bleker http://orcid.org/0000-0003-2949-9784
Susanne R de Rooij http://orcid.org/0000-0001-7382-5749
Rebecca C Painter http://orcid.org/0000-0001-9336-6033
Anita CJ Ravelli http://orcid.org/0000-0002-3447-8286
Tessa J Roseboom http://orcid.org/0000-0003-0564-5994

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
