## [Reviewer comments · BMJ Open]

ARTICLE DETAILS

TITLE (PROVISIONAL)	Cohort Profile: The Dutch famine birth cohort (DFBC), a prospective birth cohort study in the Netherlands
AUTHORS	Bleker, Laura; de Rooij, Susanne; Painter, Rebecca; Ravelli, Anita C.J.; Roseboom, Tessa

VERSION 1 – REVIEW

REVIEWER	Kirsten Donald, Jess Ringshaw, Sashmi Moodley University of Cape Town, South Africa
REVIEW RETURNED	18-Jul-2020

GENERAL COMMENTS	Reviewers: Kirsty Donald, Jessica Ringshaw, Sashmi Moodley 1) Rationale Is the article rationale well developed? Sections: Rationale, fetal origins of health and disease, historical events leading to Dutch famine, consequences to Dutch prenatal undernutrition General Comment: The study presents a unique opportunity to study the role of acute maternal undernutrition on long-term health outcomes in adult offspring. The well-documented participant medical records and the longitudinal follow-up design constitutes a strong and valid methodology for meaningful research. However, the review could benefit from including more information regarding how previous research on nutrition has informed the study question and aims and, in turn, how the specific design of this study may uniquely contribute to existing literature and knowledge on nutrition and health. Additionally, the rationale needs a stronger argument for the necessity, significance and benefits of this type of research in various contexts. Structure: The literature review should be structured in a manner that builds a strengthening argument to motivate the study. I would suggest moving the rationale to the end (rather than it being the first paragraph) of the literature review as a culminating conclusion for the purpose of the research. The “fetal origins of health and disease” section could be restructured to allow for a more logical and less repetitive flow of ideas. Currently, it oscillates between discussing the link between birth weight versus nutrition on health outcomes without concisely
---

	linking the concepts. Additionally, there is limited information regarding the scientific basis for the discussed relationships between maternal nutrition, birth weight, and health outcomes later in life. I would suggest discussing the “fetal origins of health and disease” in following way/order: 1) How does maternal nutrition affect gestational development? 2) What is the link between poor maternal nutrition and low birth weight? 3) How do the consequences of prenatal malnutrition impact/determine health outcomes later in life 4) How doe this tie together? What theories and proposed mechanisms explain these relationships? The study covers a range of medical conditions encompassing cardiovascular and metabolic health, mental health, organ function, genetics and epigenetics and brain imaging. However, the literature review does not discuss very much literature on nutrition that specifically relates to these health outcomes. Although the study is designed to be exploratory, the rationale could still be strengthened by identifying what is known about the relationship between nutrition and these health outcomes of interest , and what is yet to be investigated. This will provide more guidance and support for the study question as well as the hypotheses and expectations. The literature review draws on several other studies on the Dutch famine which provides useful context and comparative data/results, especially with regard to birth weight and mental health. However, these studies could be used to build a more cohesive argument that is tied together conceptually rather than in accordance with a historical timeline. Additionally, it may be useful to consider research on malnourishment in other contexts as well in order to cover more outcomes in greater depth. This would provide the reader with a better idea of what gaps in the literature exist, and how consistent these findings are. 2) If so, is it adequately referenced, representing the literature in this field? General: The references in the section on “fetal origins of health and disease” are slightly outdated. Only four of the references are recent (2011, 2013, 2017 and 2018) while the others tend to be from the early 2000s and 1990s (1996, 2004, 2008, 1993, 1989, 1986, 1998, 2007, 1993). Given that this initial section is the intended to provide a scientific basis for the relationship between pre-natal nutrition and health outcomes, it may be worthwhile to consider some more recent research. Additionally, as mentioned with regard to the rationale, this section could be strengthened by including more a more detailed scientific rationale for the study with regard to the specific health outcomes of interest. For example, MK Georgieff has published numerous papers on nutrition and brain development which could be very useful in explaining the mechanisms relevant to the brain imaging outcomes of the study. There is similar research and several theories that could be relevant to other outcomes of interest. At the end of the section on “fetal origins of health and disease”, the historical context of the Dutch famine is provided but not referenced at all.
--	--

	Under 'Consequences of prenatal undernutrition', there are several references to research on the Dutch famine and birth outcomes as well as mental health outcomes. However, there are not many references to studies that have focused on the other health outcomes of interest to this study (e.g. cardiovascular health, metabolic health, brain imaging etc.). Additionally, it may be worth considering how this section could benefit from a wider source of references that examine the same relationships in similar contexts. For example, much research has been done on famine and malnourishment in Africa that could still be relevant for comparative purposes. Technical: Many of the in-text references appear too late in text, only being cited at the very end of the paragraph (e.g. first paragraph of 'Consequences of Prenatal Undernutrition'; page 8, line 12-26). In text references should appear the first time an idea/concept/finding is presented. Formatting error on page 7, line 31. 3) Is the study question well-articulated? The study aim to investigate the effects of prenatal exposure to the Dutch famine on adult health is clear throughout. However, there is no section that concisely articulates the aims, objectives and hypotheses in a sequential manner. 4) Is the methodology clear and adequately described (as in would someone be able to replicate this analysis from this description?) The methodology is clearly described in detail and the study is replicable. Methodological strengths: Large sample size, well-documented historical information (famine restriction data), retention of medical records, longitudinal design, control group, comprehensive medical follow-ups (a broad range of medical outcomes investigated). Methodological Limitations: 1) Difficulty accounting for the impact of post-natal factors on health outcomes limits the ability to establish cause-effect relationships. 2) the control group constitutes individuals born before or after the famine. The individuals born before the famine (1 year prior) may not be a useful comparative group given that they would have still lived through the famine. Malnourishment in early child development (especially first 1000 days) is known to be detrimental to development and health outcomes. It seems like young children were somewhat protected from the famine (caloric restriction never dipped below 1000) although they were still not being adequately nourished. 5) Are the results valid and important The results revealed several significant findings indicating that exposure to acute malnutrition in gestation is associated with numerous poor health outcomes in adult offspring. This validates the need for interventions that target nutrition in pregnant mothers. Additionally, the effects were dependent on timing during gestation which provides support for research that emphasizes the role of timing, dose and duration of nutrition in development. The
--	---

scientific basis for this was discussed well. However, there are other more specific theories that could be relevant here with regard to specific outcomes as the study progresses. For example, the Regionalization Hypothesis (Georgieff et al., 2018) proposes that brain development may be nonhomogenous and regional in nature whereby different brain regions and neural processes have varying developmental trajectories. Therefore, the impact of a nutrient deficiency on the brain depends on the time at which it occurs during the developmental trajectory as well as the brain region's need for the nutrient at the time. This research on the Dutch famine is in alignment with this hypothesis although, in this article, it speaks to it more broadly with regard to the organogenesis of different organ systems. This is a significant finding with clinical relevance as it provides evidence for varying nutritional needs during different stages of pregnancy.

Lastly, the results indicated that exposure to famine was associated with poor long-term health outcomes, but this was relatively independent of birth weight. This is important as most research thus far has focused on birth weight as a proxy for longitudinal health outcomes, with low birth weight being an indicator of a poorer prognosis. This research suggests that birth weight is not necessarily predictive of adult health outcomes and, therefore, the risk for famine-exposed neonates extends beyond babies that are underweight. This highlights the importance of following up on all children who are born to undernourished mothers even if the infant birth weight is normal. This contradicts the traditional classification of babies as "high-risk" based on low birth weight, and extends it to all children exposed to acute malnutrition in utero.

The importance of these findings indicated above is not discussed in the article.

6) Does the discussion reflect the results?

The results are significant and of great importance in both research and clinical contexts. However, the articles merely states the results and there is no discussion regarding the significance of such or importance of such findings as indicated above.

7) Limitations adequately discussed?

The following limitations were adequately discussed: Genetic confounders, other incidental consequences of the famine on pregnant mothers (e.g. psychological/physiological stress), selective conception and survival, selective attrition.

Limitations not discussed: The study has not adequately discussed the role of other potential causes for poor health outcomes in adults such as post-natal SES (including adult life), Lifestyle and nutrition. It has also not discussed the role of the famine and poor nutrition in early child development within the control group born within 1 year before the famine.

8) Worth publishing?

	This article is worth publishing as it is a unique study with very specific insight into the impact on acute malnutrition in utero on longitudinal health outcomes in adults. The findings are also valid and significant to both research and clinical contexts, providing novel knowledge. Furthermore, the study is ongoing and demonstrates great exploratory potential. 9) Things that could be improved (detailed list of queries or suggestions)  - The literature review and rationale needs to present a stronger argument to motivate the study. - The literature review needs a stronger scientific basis as well as more up-to-date references. - There is no discussion of the results and their significance or implications. - There are a few further limitations that could be acknowledged and discussed.
--	---

REVIEWER	Susanne Grylka-Baesclin Zurich University of Applied Sciences School of Health Professions, Institute of Midwifery Switzerland
REVIEW RETURNED	06-Oct-2020

GENERAL COMMENTS	Thank you very much for giving me the opportunity to review this article. The Dutch famine cohort study is an interesting project and was a unique opportunity to study effects of undernutrition during pregnancy on later offspring. It seems that the first wave of the cohort was conducted over 25 years ago and that many articles were published about this cohort during this quarter of a century. I also understand that a sixth wave is currently ongoing. Whereas the title informs that the article provides a profile of the Dutch famine cohort study, the manuscript itself does not include a clear aim. Furthermore, at several parts, the structure of the article remains unclear. Abstract There are some concerns about the structure of the abstract:  - Purpose: The study design would not belong to the purpose. The purpose of the Dutch famine cohort study is described but not the purpose of this article. - Findings to date, lines 22-38: Most information in this finding summary seems to belong to the methods. It remains unclear, why there is no methods section in the abstract. Introduction The introduction is rather long and includes foetal origin of health and diseases, historical aspects of the Dutch famine and consequences of prenatal undernutrition. The chapter is well written, however:  - There is no clear structure in this chapter, it jumps from medical to historical and back to medical aspects. - It does not provide a clear rational an lead to an aim for this article. - It includes information about how the Dutch famine cohort was built, what would rather belong to the methods section. Cohort description
---

	In this chapter, there are some concerns about redundancies and the demarcation with the findings chapter. It is unclear, whether the cohort description is a methods chapter or a mix between methods and characteristics of participants.  - It would be better to put all information about inclusion and exclusion criteria together. This would avoid to repeat several times the “live born term singletons” and on page 11, line 50, the sentence “Twins and stillbirths were excluded” would not be necessary. - Table 1 about maternal and birth characteristics is referenced in the cohort description chapter as well as in the result chapter. This seems unusual and is a further indication of an unclear structure. - Page 16, line 25: it was already mentioned at the beginning of the paragraph that a sixth wave is in progress. Findings to date The summary of the findings to date is interesting. Is this the first publication which summarise all the findings? If yes, this would be a good rationale for the paper. If no, it should be . Strengths and limitations The strengths and limitations are very elaborated and good. However, a real discussion is missing. The authors mention the Dutch hunger winter family study with a slightly different approach. The reader would be interested, whether results of the studies were similar or if there were major differences and why differences might be. Additionally, the article finishes with a weakness of the cohort. There is no conclusion and no outlook. The paper is not a “round thing”. Literature The authors use a huge amount of literature for this article. However, there are many old and only very few new references. I understand that this is also because of the publications about Dutch famine cohort study. However, I wonder, whether for the epidemiological studies about consequences of low birth weight (references 1-7) there are not also many recent publications, which could be referenced.  - The reviewer provided a marked copy with additional comments. Please contact the publisher for full details.
--	---

REVIEWER	Stefania Papatheodorou Harvard School of Public Health, USA
REVIEW RETURNED	08-Oct-2020

GENERAL COMMENTS	Re: Cohort profile: The Dutch famine birth cohort (DFBC), a prospective birth cohort study in the Netherlands. Thank you for giving me the opportunity to review this paper. The study itself is very interesting and the paper is well written. I have some suggestions that will make the paper shorter, easier to read, and hopefully add some elements that will be useful for the readers of this paper. 1. The overall goal of this paper is not clear to me. Is it a summary so other researchers know what is available and propose new questions? Is it going to be used as a reference for future publication using this cohort? It would be useful to add a phrase to be clear about what the goal of this paper is.
--

	2. The historical events are indeed extremely interesting, and they need to be described for the readers to understand what happened, but they occupy a lot of space. If only the key events are described, this section will become a paragraph or less. 3. The same holds for the early studies of the Dutch famine. They are interesting but they take too much space and their epidemiological methods may not be relevant, since we have modern tools to address the issues of confounding and analysis of time-varying covariates. 4. Page 11/line 22: The authors mention that the study included only full-term pregnancies. It would be useful to describe the rationale for excluding the preterm deliveries since nutritional status has been associated with preterm birth. Also, in the next sentences, the authors are describing the way that gestational age has been calculated. During the 1940s the first prenatal visit could happen very late during pregnancy so how accurate calculation of gestational age can be? If all these data came from handwritten notes and were measured in 1940, the potential for measurement errors needs to be discussed. 5. The comparison that has been performed in this cohort is interesting. The cohort consists of 821 participants that were exposed to famine prenatally, 764 infants born before, and 829 born after. How was the fact that a baby born before the famine was exposed to famine were exposed after, at a very early age? Was that considered? This issue comes up again on Page 12/ line 33. 6. Page 12/line 24: Since the authors have examined the windows of exposure, were they able to examine the effects of exposure to one window accounting for other windows of exposure? I believe that since the design is so interesting, it would be a great opportunity to discuss some of the analytical challenges the authors have faced and how they have approached them. 7. Page 13/line 12: This confirms what I said earlier as a comment on pregnancy dating. 8. Page 13/line 56: Here, for the first time, the authors report that the WG hospital facilitates deliveries for unmarried women and women with poor housing. So is this a special sub-sample of the total population? 9. There is no referral to the Bioethics committee approval which I sure exist. 8. Page 13/line 24: The calculation of maternal weight gain is not clear to me. According to what was said before, the first prenatal visit could be very close to the beginning of the third trimester. 9. The tables can be more compact; they are hard to follow as they are now. 10. 9. Page 15/line 37: The fact that there was almost 40% “lost to follow up” needs to be highlighted in terms of the potential for bias. The authors mention it at the end of the paper, but the concept of competing risks should be described in more detail. 10. Page 46/line 44: The abbreviation MRC has not been defined before. Also, the authors need to explain why they used this study. 11. Page 17: Table 2 describes very clearly the data collected in every wave. A graph would be informative here as well (displaying linear time). Also, it is very clear that there was a significant loss to follow up ending in 44% of the original sample size in wave III. 12. Page 23/line 15: The authors should discuss the issue of confounding by birth weight to a greater extend. If famine is the exposure and let’s say metabolic syndrome is the outcome, birthweight could be considered as an intermediate and therefore
--	--

	should not be adjusted for. Another possible exploration would be mediation by birthweight. In general, it would be more useful for the authors to describe the methodological challenges they faced when analyzing their data which could lead to other interesting explorations to the same or different research questions.
--	--

VERSION 1 – AUTHOR RESPONSE

Reviewers' Comments to Author:

Reviewer: 1

1) Rationale

Is the article rationale well developed?

Sections: Rationale, fetal origins of health and disease, historical events leading to Dutch famine, consequences to Dutch prenatal undernutrition

General Comment:

The study presents a unique opportunity to study the role of acute maternal undernutrition on longterm health outcomes in adult offspring. The well-documented participant medical records and the longitudinal follow-up design constitutes a strong and valid methodology for meaningful research. However, the review could benefit from including more information regarding how previous research on nutrition has informed the study question and aims and, in turn, how the specific design of this study may uniquely contribute to existing literature and knowledge on nutrition and health. Additionally, the rationale needs a stronger argument for the necessity, significance and benefits of this type of research in various contexts.

Authors' response: We have added several lines explaining the unique quasi-experimental situation provided by the Dutch famine, the establishment of the DFBC following early work on the fetal origins hypothesis (p5 towards the end, p6, p11), the design of the DFBC (p6) and the importance of studies on undernutrition during pregnancy and long term impact (Study rationale (p5), Implications section (p37)).

Structure:

The literature review should be structured in a manner that builds a strengthening argument to motivate the study. I would suggest moving the rationale to the end (rather than it being the first paragraph) of the literature review as a culminating conclusion for the purpose of the research. The “fetal origins of health and disease” section could be restructured to allow for a more logical and less repetitive flow of ideas. Currently, it oscillates between discussing the link between birth weight versus nutrition on health outcomes without concisely linking the concepts. Additionally, there is limited information regarding the scientific basis for the discussed relationships between maternal nutrition, birth weight, and health outcomes later in life. I would suggest discussing the “fetal origins of health and disease” in following way/order: 1) How does maternal nutrition affect gestational development? 2) What is the link between poor maternal nutrition and low birth weight?

3) How do the consequences of prenatal malnutrition impact/determine health outcomes later in life 4) How does this tie together? What theories and proposed mechanisms explain these relationships?

The study covers a range of medical conditions encompassing cardiovascular and metabolic health, mental health, organ function, genetics and epigenetics and brain imaging. However, the literature review does not discuss very much literature on nutrition that specifically relates to these health outcomes. Although the study is designed to be exploratory, the rationale could still be strengthened by identifying what is known about the relationship between nutrition and these health outcomes of interest, and what is yet to be investigated. This will provide more guidance and support for the study question as well as the hypotheses and expectations.

The literature review draws on several other studies on the Dutch famine which provides useful context and comparative data/results, especially with regard to birth weight and mental health. However, these studies could be used to build a more cohesive argument that is tied together conceptually rather than in accordance with a historical timeline. Additionally, it may be useful to consider research on malnourishment in other contexts as well in order to cover more outcomes in greater depth. This would provide the reader with a better idea of what gaps in the literature exist, and how consistent these findings are.

Authors' response: We thank the reviewers for their advice. We agree that building up the literature review in this way would provide a nice and coherent story of the fetal origins hypothesis and the role of nutrition. However, the current manuscript is a Cohort profile and therefore we have chosen to give a distinct description of the literature that has formed the basis of the establishment of the cohort. We feel that such an extensive review of the literature would not fit the purpose of the manuscript, which is to describe the Dutch famine birth cohort.

2) If so, is it adequately referenced, representing the literature in this field?

General:

The references in the section on "fetal origins of health and disease" are slightly outdated. Only four of the references are recent (2011, 2013, 2017 and 2018) while the others tend to be from the early 2000s and 1990s (1996, 2004, 2008, 1993, 1989, 1986, 1998, 2007, 1993). Given that this initial section is intended to provide a scientific basis for the relationship between pre-natal nutrition and health outcomes, it may be worthwhile to consider some more recent research. Additionally, as mentioned with regard to the rationale, this section could be strengthened by including more a more detailed scientific rationale for the study with regard to the specific health outcomes of interest. For example, MK Georgieff has published numerous papers on nutrition and brain development which could be very useful in explaining the mechanisms relevant to the brain imaging

outcomes of the study. There is similar research and several theories that could be relevant to other outcomes of interest.

At the end of the section on “fetal origins of health and disease”, the historical context of the Dutch famine is provided but not referenced at all.

Under ‘Consequences of prenatal undernutrition’, there are several references to research on the Dutch famine and birth outcomes as well as mental health outcomes. However, there are not many references to studies that have focused on the other health outcomes of interest to this study (e.g. cardiovascular health, metabolic health, brain imaging etc.). Additionally, it may be worth considering how this section could benefit from a wider source of references that examine the same relationships in similar contexts. For example, much research has been done on famine and malnourishment in Africa that could still be relevant for comparative purposes.

Technical:

Many of the in-text references appear too late in text, only being cited at the very end of the paragraph (e.g. first paragraph of ‘Consequences of Prenatal Undernutrition’; page 8, line 12-26). In-text references should appear the first time an idea/concept/finding is presented.

Formatting error on page 7, line 31.

Authors’ response: We very much agree with the reviewers that some of the literature was a bit outdated and some references were missing, We did a thorough job of going through all the literature we cited and updated references as well as added references where needed.

Again, we feel that it is not the aim of the present manuscript to provide such an elaborate review of the literature and/or place the findings in the cohort in the context of the broader literature. We did add a paragraph on the alignment of findings in our cohort to that in other cohorts investigating consequences of famine (p37).

3) Is the study question well-articulated?

The study aim to investigate the effects of prenatal exposure to the Dutch famine on adult health is clear throughout. However, there is no section that concisely articulates the aims, objectives and hypotheses in a sequential manner.

Authors’ response: Thank you. We have now added the aim and hypothesis (Study rationale, p5).

4) Is the methodology clear and adequately described (as in would someone be able to replicate this analysis from this description?)

The methodology is clearly described in detail and the study is replicable.

Methodological strengths: Large sample size, well-documented historical information (famine

restriction data), retention of medical records, longitudinal design, control group, comprehensive medical follow-ups (a broad range of medical outcomes investigated).

Methodological Limitations: 1) Difficulty accounting for the impact of post-natal factors on health outcomes limits the ability to establish cause-effect relationships. 2) the control group constitutes individuals born before or after the famine. The individuals born before the famine (1 year prior) may not be a useful comparative group given that they would have still lived through the famine.

Malnourishment in early child development (especially first 1000 days) is known to be detrimental to development and health outcomes. It seems like young children were somewhat protected from the famine (caloric restriction never dipped below 1000) although they were still not being adequately nourished.

Authors' response: Yes, agreed, these points are discussed in the manuscript.

5) Are the results valid and important

The results revealed several significant findings indicating that exposure to acute malnutrition in gestation is associated with numerous poor health outcomes in adult offspring. This validates the need for interventions that target nutrition in pregnant mothers.

Additionally, the effects were dependent on timing during gestation which provides support for research that emphasizes the the role of timing, dose and duration of nutrition in development. The scientific basis for this was discussed well. However, there are other more specific theories that could be relevant here with regard to specific outcomes as the study progresses. For example, the Regionalization Hypothesis (Georgieff et al., 2018) proposes that brain development may be nonhomogenous and regional in nature whereby different brain regions and neural processes have varying developmental trajectories. Therefore, the impact of a nutrient deficiency on the brain depends on the time at which it occurs during the developmental trajectory as well as the brain region's need for the nutrient at the time. This research on the Dutch famine is in alignment with this hypothesis although, in this article, it speaks to it more broadly with regard to the organogenesis of different organ systems. This is a significant finding with clinical relevance as it provides evidence for varying nutritional needs during different stages of pregnancy.

Lastly, the results indicated that exposure to famine was associated with poor long-term health outcomes, but this was relatively independent of birth weight. This is important as most research thus far has focused on birth weight as a proxy for longitudinal health outcomes, with low birth weight being an indicator of a poorer prognosis. This research suggests that birth weight is not

necessarily predictive of adult health outcomes and, therefore, the risk for famine-exposed neonates extends beyond babies that are underweight. This highlights the importance of following up on all children who are born to undernourished mothers even if the infant birth weight is normal. This contradicts the traditional classification of babies as “high-risk” based on low birth weight, and extends it to all children exposed to acute malnutrition in utero.

The importance of these findings indicated above is not discussed in the article.

Authors' response: Yes, we completely agree with the fact that the finding of our results being independent of birth weight is highly significant. It is described as one of the three main conclusions of the studies in the cohort (p28). The Regionalization Hypothesis is highly interesting, but we feel that the brain outcomes we showed in our cohort so far are rather general (whole brain volume, Brainage and brain perfusion) and not region specific.

6) Does the discussion reflect the results?

The results are significant and of great importance in both research and clinical contexts. However, the articles merely states the results and there is no discussion regarding the significance of such or importance of such findings as indicated above.

Authors' response: We disagree with this observation. The section entitled 'Findings to date' provides a discussion of the most important results of the Dutch famine birth cohort study. It contains a description of the three points that we feel are the most significant to have come out of our studies.

7) Limitations adequately discussed?

The following limitations were adequately discussed: Genetic confounders, other incidental consequences of the famine on pregnant mothers (e.g. psychological/physiological stress), selective conception and survival, selective attrition.

Limitations not discussed: The study has not adequately discussed the role of other potential causes for poor health outcomes in adults such as post-natal SES (including adult life), Lifestyle and nutrition. It has also not discussed the role of the famine and poor nutrition in early child development within the control group born within 1 year before the famine.

Authors' response:

The reviewers are correct, we did not discuss potential postnatal confounding. We have now added this as last point to the Strengths and Limitations section (p36).

Regarding the control group of people who were born before the famine, as the reviewers have also noted, infants were relatively protected from the famine being entitled to receiving higher rations during the famine. Also, the outcomes in this group in later life were mostly highly comparable to those in the group of people conceived after the famine. We have added the latter to the Strengths and Limitations section (p34, top).

8) Worth publishing?

This article is worth publishing as it is a unique study with very specific insight into the impact on acute malnutrition in utero on longitudinal health outcomes in adults. The findings are also valid and significant to both research and clinical contexts, providing novel knowledge. Furthermore, the study is ongoing and demonstrates great exploratory potential.

9) Things that could be improved (detailed list of queries or suggestions)

- The literature review and rationale needs to present a stronger argument to motivate the study.
- The literature review needs a stronger scientific basis as well as more up-to-date references.
- There is no discussion of the results and their significance or implications.
- There are a few further limitations that could be acknowledged and discussed.

Authors' response: We have responded to all of these points in the above.

Reviewer: 2

Thank you very much for giving me the opportunity to review this article. The Dutch famine cohort study is an interesting project and was a unique opportunity to study effects of undernutrition during pregnancy on later offspring. It seems that the first wave of the cohort was conducted over 25 years ago and that many articles were published about this cohort during this quarter of a century. I also understand that a sixth wave is currently ongoing. Whereas the title informs that the article provides a profile of the Dutch famine cohort study, the manuscript itself does not include a clear aim. Furthermore, at several parts, the structure of the article remains unclear.

Authors' response: Thank you. Actually, during the time of the reviewing process, the sixth wave has been completed, and the seventh is planned!

We have added the aim of the study to the Rationale section on page 5. Overall, we have tried to improve the structure and readability of the manuscript.

Abstract

There are some concerns about the structure of the abstract:

- Purpose: The study design would not belong to the purpose. The purpose of the Dutch famine cohort study is described but not the purpose of this article.

Authors' response: We have removed the design from the purpose section in the abstract. We have followed the guidelines of the BMJ Open for Cohort profiles stating that under Purpose it has to be described why the cohort was set up (so not the purpose of the article).

- Findings to date, lines 22-38: Most information in this finding summary seems to belong to the methods. It remains unclear, why there is no methods section in the abstract.

Authors' response: Here again, we have followed the guidelines of the BMJ Open for a cohort profile abstract.

Introduction

The introduction is rather long and includes foetal origin of health and diseases, historical aspects of the Dutch famine and consequences of prenatal undernutrition. The chapter is well written, however:

- There is no clear structure in this chapter, it jumps from medical to historical and back to medical aspects.
- It does not provide a clear rationale and lead to an aim for this article.
- It includes information about how the Dutch famine cohort was built, what would rather belong to the methods section.

Authors' response: Thanks for pointing this out. We have rewritten this chapter. We hope it is more clear now.

Cohort description

In this chapter, there are some concerns about redundancies and the demarcation with the findings chapter. It is unclear, whether the cohort description is a methods chapter or a mix between methods and characteristics of participants.

- It would be better to put all information about inclusion and exclusion criteria together. This would avoid to repeat several times the "live born term singletons" and on page 11, line 50, the sentence "Twins and stillbirths were excluded" would not be necessary.
- Table 1 about maternal and birth characteristics is referenced in the cohort description chapter as well as in the result chapter. This seems unusual and is a further indication of an unclear structure.
- Page 16, line 25: it was already mentioned at the beginning of the paragraph that a sixth wave is in progress.

Authors' response: . We have removed the redundancy in the information on eligibility. Table I is indeed referenced in both methods and results. Maternal and birth characteristics were both used as criteria for inclusion in the cohort, but also contain information on the differences in for example birth weight between the different exposure groups, which is described in the Results section. We have updated the information on wave VI and removed the double information.

Findings to date

The summary of the findings to date is interesting. Is this the first publication which summarise all the findings? If yes, this would be a good rationale for the paper. If no, it should be .

Authors' response: This is not the first paper describing joint outcomes of the cohort. However, this is the first Cohort Profile summarizing the set-up, design, and overview of results so far. It has also been submitted as a Cohort Profile, which is a specific sort of papers published in BMJ Open.

Strengths and limitations

The strengths and limitations are very elaborated and good. However, a real discussion is missing. The authors mention the Dutch hunger winter family study with a slightly different approach. The reader would be interested, whether results of the studies were similar or if there were major differences and why differences might be.

Additionally, the article finishes with a weakness of the cohort. There is no conclusion and no outlook. The paper is not a "round thing".

Authors' response: Thanks for pointing this out, we agree. We have now added a line to the paragraph about the Dutch hunger winter family study, saying that findings between the two cohorts are strikingly similar (p35). Also, we have added two sections to the end of the paper entitled 'Similar findings in different settings' and 'Implications', which we feel make a better completion of the paper.

Literature

The authors use a huge amount of literature for this article. However, there are many old and only very few new references. I understand that this is also because of the publications about Dutch famine cohort study. However, I wonder, whether for the epidemiological studies about consequences of low birth weight (references 1-7) there are not also many recent publications, which could be referenced.

Authors' response: We very much agree with the reviewer that some of the literature was outdated. We did a thorough job of going through all the literature we cited and updated references where we felt this was appropriate.

Reviewer: 3

Thank you for giving me the opportunity to review this paper. The study itself is very interesting and the paper is well written. I have some suggestions that will make the paper shorter, easier to read, and hopefully add some elements that will be useful for the readers of this paper.

Authors' response: We thank the reviewer for the compliments. We have made the paper shorter and hopefully easier to read.

1. The overall goal of this paper is not clear to me. Is it a summary so other researchers know what is available and propose new questions? Is it going to be used as a reference for future publication using this cohort? It would be useful to add a phrase to be clear about what the goal of this paper is.

Authors' response: The paper is a Cohort profile paper, as described in the BMJ Open: "Cohort profiles should describe the rationale for a cohort's creation, its methods, baseline data and its future plans. Cohorts described should be long-term, prospective projects and not time-limited cohorts established to answer a small number of specific research questions."

2. The historical events are indeed extremely interesting, and they need to be described for the readers to understand what happened, but they occupy a lot of space. If only the key events are described, this section will become a paragraph or less.

Authors' response: We understand and have shortened the text. It now consists of one paragraph.

3. The same holds for the early studies of the Dutch famine. They are interesting but they take too much space and their epidemiological methods may not be relevant, since we have modern tools to address the issues of confounding and analysis of time-varying covariates.

Authors' response: We have shortened this bit as well.

4. Page 11/line 22: The authors mention that the study included only full-term pregnancies. It would be useful to describe the rationale for excluding the preterm deliveries since nutritional status has been associated with preterm birth. Also, in the next sentences, the authors are describing the way that gestational age has been calculated. During the 1940s the first prenatal visit could happen very late during pregnancy so how accurate calculation of gestational age can be? If all these data came from handwritten notes and were measured in 1940, the potential for measurement errors needs to be discussed.

Authors' response: Preterm deliveries can have short and longterm consequences for health in itself. If we would have included preterm babies, it would have been difficult to attribute potential effects of famine to famine. We have added this information to the text (p12). Indeed, there was potential for measurement errors. However, we do not expect these to be different for the different groups.

5. The comparison that has been performed in this cohort is interesting. The cohort consists of 821 participants that were exposed to famine prenatally, 764 infants born before, and 829 born after. How was the fact that a baby born before the famine was exposed to famine were exposed after, at a very early age? Was that considered? This issue comes up again on Page 12/ line 33.

Authors' response: Children who were born before the famine were indeed exposed to the famine as infants. However, infants were relatively protected from the famine. They were entitled to greater amounts of food than the rest of the population. We have described these circumstances on page 8 (at the top). Also, the later life health of people born before the famine was comparable to the health of people who were conceived after the famine, whereas differences between these groups would be expected if the famine exposure during infancy had a programming effect. We have added this comment to the Strengths and Limitations section (p36, top).

6. Page 12/line 24: Since the authors have examined the windows of exposure, were they able to examine the effects of exposure to one window accounting for other windows of exposure? I believe that since the design is so interesting, it would be a great opportunity to discuss some of the analytical challenges the authors have faced and how they have approached them.

Authors' response: In all of our studies we have compared those exposed to famine in early, mid or late gestation to those prenatally unexposed to famine. We have mostly analyzed this in standard

regression models with early, mid and late exposure as dummy variables. We have added this information to the Section Measurements (p17).

7. Page 13/line 12: This confirms what I said earlier as a comment on pregnancy dating.

Authors' response: We are not completely sure what the reviewer means here, but hopefully our explanation to point 4 is explanatory.

8. Page 13/line 56: Here, for the first time, the authors report that the WG hospital facilitates deliveries for unmarried women and women with poor housing. So is this a special sub-sample of the total population?

Authors' response: Most patients visiting the WG hospital came from a low to middle class background. There is not a lot of information about the referral pattern during the famine. It is likely that women of all classes at that time attended the hospital as conditions were poor for all people living in Amsterdam at that moment. We understand that this information may be confusing, we have removed it from the text.

9. There is no referral to the Bioethics committee approval which I sure exist.

Authors' response: Yes, it does indeed, we have now added the exact name of our ethical committee as well as the numbers referencing the ethical approval documents (p17).

8. Page 13/line 24: The calculation of maternal weight gain is not clear to me. According to what was said before, the first prenatal visit could be very close to the beginning of the third trimester.

Authors' response: Sorry about the confusion, the prenatal visit described here is the last prenatal visit before birth. We have added text to explain this in more detail.

9. The tables can be more compact; they are hard to follow as they are now.

Authors' response: We are sorry to hear this. We have tried to make them as compact as possible and as there are five different study groups, we can think of no way to make them more compact than they are right now. If the reviewers has any suggestions for this, we are of course open to these.

9. Page 15/line 37: The fact that there was almost 40% "lost to follow up" needs to be highlighted in terms of the potential for bias. The authors mention it at the end of the paper, but the concept of competing risks should be described in more detail.

Authors' response: We completely agree that selective attrition and participation is a limitation to our study. We have described this in detail in the Limitations section (p36) and also as one of the most important limitations as a bullet point following the abstract.

10. Page 46/line 44: The abbreviation MRC has not been defined before. Also, the authors need to explain why they used this study.

Authors' response: We have now written out Medical Research Council (p11) and also explained why we have used the results from these studies (p16).

11. Page 17: Table 2 describes very clearly the data collected in every wave. A graph would be informative here as well (displaying linear time). Also, it is very clear that there was a significant loss to follow up ending in 44% of the original sample size in wave III.

Authors' response: We have made a figure displaying the Waves, years of waves and mean ages at time of the waves. Yes, there was loss to follow-up, probably to do with the increasing age as well as people moving, emigrating or dying.

12. Page 23/line 15: The authors should discuss the issue of confounding by birth weight to a greater extend. If famine is the exposure and let's say metabolic syndrome is the outcome, birthweight could be considered as an intermediate and therefore should not be adjusted for. Another possible exploration would be mediation by birthweight. In general, it would be more useful for the authors to describe the methodological challenges they faced when analyzing their data which could lead to other interesting explorations to the same or different research questions.

Authors' response: All of the analyses we performed were initially not adjusted for birth weight and in additional analyses were adjusted for birth weight. Adjusting for birth weight did not change the results that were found without adjusting for birth weight. Most results that were found were in the group exposed to famine in early gestation and this group had birth weights that were similar to that of the control groups, so it makes sense that effects were not confounded or mediated by birth weight. We discuss this quite extensively in our point 3 of Findings to date (p28).

VERSION 2 – REVIEW

REVIEWER	Susanne Grylka-Baesclin Zurich University of Applied Sciences, Switzerland
REVIEW RETURNED	05-Jan-2021

GENERAL COMMENTS	Thank you for thoroughly revising this manuscript. I agree with the changes.
--

REVIEWER	Stefania Papatheodorou Harvard TH Chan School of Public Health
REVIEW RETURNED	05-Jan-2021

GENERAL COMMENTS	I believe that the authors addressed my and other reviewer's comments adequately. There are some minor issues to be addressed before publication: 1. There are minor formatting issues throughout the text (e.g page 8 line 30 is missing a bracket, 2 full stops on page 12 line 31) 2. Page 10 line 14: the authors mention a 13 week period and in line 19 they define 16 week periods.
---